# Acute kidney injury in Ugandan children with severe malaria is associated with long-term behavioral problems

**Meredith R. Hickson[1]☯, Andrea L. Conroy[2]☯*, Paul Bangirana[3], Robert O. Opoka[4], Richard Idro[4,5], John M. Ssenkusu[6], Chandy C. John[2]**

**1** Children's Hospital of Philadelphia, Philadelphia, Pennsylvania, United States of America, **2** Ryan White Center for Pediatric Infectious Disease and Global Health, Indiana University School of Medicine, Indiana, United States of America, **3** Department of Psychiatry, Makerere University of School of Medicine, Kampala, Uganda, **4** Department of Paediatrics and Child Health, Makerere University School of Medicine, Kampala, Uganda, **5** Centre of Tropical Medicine and Global Health, University of Oxford, Oxford, United Kingdom, **6** Department of Epidemiology and Biostatistics, Makerere University School of Public Health, Kampala, Uganda

☯ These authors contributed equally to this work.
* conroya@iu.edu

**Data Availability Statement:** All relevant data are within the paper and its Supporting information files.

## Abstract

### Background

Acute kidney injury (AKI) is a risk factor for neurocognitive impairment in severe malaria (SM), but the impact of AKI on long-term behavioral outcomes following SM is unknown.

### Methods

We conducted a prospective study on behavioral outcomes of Ugandan children 1.5 to 12 years of age with two forms of severe malaria, cerebral malaria (CM, n = 226) or severe malarial anemia (SMA, n = 214), and healthy community children (CC, n = 173). AKI was defined as a 50% increase in creatinine from estimated baseline. Behavior and executive function were assessed at baseline and 6, 12, and 24 months later using the Child Behavior Checklist and Behavior Rating Inventory of Executive Function, respectively. Age-adjusted z-scores were computed for each domain based on CC scores. The association between AKI and behavioral outcomes was evaluated across all time points using linear mixed effect models, adjusting for sociodemographic variables and disease severity.

### Results

AKI was present in 33.2% of children with CM or SMA at baseline. Children ≥6 years of age with CM or SMA who had AKI on admission had worse scores in socio-emotional function in externalizing behaviors (Beta (95% CI), 0.52 (0.20, 0.85), p = 0.001), global executive function (0.48 (0.15, 0.82), p = 0.005) and behavioral regulation (0.66 (0.32, 1.01), p = 0.0002) than children without AKI. There were no behavioral differences associated with AKI in children <6 years of age.

**Funding:** This work was supported by the National Institute of Neurological Disorders and Stroke (R01NS055349 to CCJ, https://www.ninds.nih.gov/) and the Fogarty International Center (D43 NS078280 to CCJ, https://www.fic.nih.gov/). The funders had no role in study design, data collection and analysis, decision to publish, or preparation of the manuscript.

**Competing interests:** The authors have declared that no competing interests exist.

## Conclusions

AKI is associated with long-term behavioral problems in children ≥6 years of age with CM or SMA, irrespective of age at study enrollment.

## Introduction

More than 200 million children worldwide are at risk of failing to meet their developmental potential [1]. Malaria due to *Plasmodium falciparum* is a major cause of mortality and neuro-developmental morbidity in sub-Saharan Africa—the region with the highest percentage of children at increased risk for developmental delay [1–4]. Severe malaria (SM) is associated with long-term impairment in multiple domains, including cognition, memory, attention, and behavior [4, 5]. Both parasite and host factors appear to contribute to the pathophysiology of SM [6]; however the mechanisms by which SM acts on the developing brain are not well understood.

Acute kidney injury (AKI) is a common, life-threatening complication of SM in both children and adults [7–13]. Recent studies in children have shown an association between AKI and neurodisability at discharge [7] and long-term neurocognitive impairment [14]. The limited availability of renal replacement therapy in malaria-endemic countries increases the risk of mortality in SM patients who present with AKI [15, 16]. This association between AKI and cognition in children with both CM and SMA may help to explain the broader impacts of SM on neurodevelopment. The effects of AKI on long-term behavioral problems in children is less well-defined, and we are unaware of any studies addressing the association of AKI with long-term behavioral disorders in any disease process.

Given the lack of data on AKI and long-term behavioral problems in children with SM, we conducted a long-term longitudinal cohort study, in which we assessed whether AKI is associated with long-term effects on behavioral outcomes in Ugandan children with SM.

## Materials and methods

### Participant enrollment

The study enrolled children between 2008 and 2013 at Mulago National Referral Hospital in Kampala, Uganda [4]. We enrolled all eligible children between the ages of 18 months and 12 years who presented to Mulago with two forms of severe malaria, cerebral malaria (CM) or severe malarial anemia (SMA). Inclusion criteria for cerebral malaria (CM) were 1) coma (Blantyre Coma Scale ≤2, Glasgow Coma Scale ≤8) with 2) *P. falciparum* on blood smear, and without 3) another identifiable cause of coma (white blood cells<10; negative lumbar puncture gram stain and culture; if hypoglycemic, no response to glucose bolus after 1 hour) [4, 17]. Inclusion criteria for severe malarial anemia (SMA) were 1) *P. falciparum* on blood smear, with 2) serum hemoglobin ≤5g/dL. Children with both CM and SMA were classified as CM. We recruited healthy community children (CC) from the same nuclear family or extended family household as a participant with CM or SMA, who were between the ages of 18 months and 12 years, and within 1 year of age of the household participant with CM or SMA.

We excluded children who lived further than 50km from Mulago and children for whom the primary caregiver reported: 1) chronic illness requiring medical care, 2) history of developmental delay, 3) history of coma (of any etiology, reported by the child's caregiver), head trauma, or cerebral palsy, or 4) history of hospitalization for malnutrition. Additional

exclusion criteria for SM without coma were: 1) impaired consciousness on exam, 2) seizure activity prior to presentation, or 3) any other clinical evidence of central nervous system disease. Additional exclusion criteria for CC were: 1) illness requiring medical care within the previous 4 weeks or 2) major medical or neurological abnormality on screening exam. Due to the high prevalence of asymptomatic malaria in Ugandan children, we treated CC with a blood smear positive for *Plasmodium* species and did not exclude them from the study [18].

### Initial clinical assessment and treatment

At enrollment, all children underwent a physical examination inclusive of the cardiovascular, pulmonary, gastrointestinal, and nervous systems. Study clinical officers collected a medical history from the participant's primary caregiver (parent or guardian). Nutritional status was evaluated with weight- and height-for-age z-scores for all children during the initial physical exam using 2007 WHO reference values (Epi Info v. 3.5.3, CDC, Atlanta GA). Per the Ugandan Ministry of Health treatment guidelines in effect at the time, children with SM received IV quinine followed by oral quinine during the initial admission and/or parenteral artemether, and oral artemisinin combination therapy after discharge. In 2012, the treatment guidelines were updated recommending intravenous artesunate as the first-line therapy for severe malaria, and intravenous artesunate was introduced in early 2013.

### Assessment of renal function

Creatinine was tested on cryopreserved plasma samples using a Beckman Coulter AU680 using the modified Jaffe method (Indiana University, Pathology Laboratory). AKI was defined using the Kidney Disease: Improving Global Outcomes (KDIGO) guidelines based on a single admission sample with premorbid creatinine estimated using the CC as previously described [14]. Briefly, we constructed a creatinine for height curve using the CC to generate a population-specific estimate of baseline creatinine and used the child's height on admission to estimate baseline creatinine using the Bedside Schwartz equation [14]. AKI was defined as a 1.5-fold increase in creatinine from baseline [19].

### Assessment of socioeconomic status and home environment

To determine participants' socioeconomic status (SES), a scored questionnaire was used at enrollment as previously described [20]. SES assessed using this is related to cognitive functioning in healthy Ugandan children [20]. In addition, levels of cognitive stimulation and emotional support provided by caregivers was assessed using the Home Observation of the Environment (HOME) questionnaire [21].

### Behavioral assessment

We selected 2 behavioral domains for inclusion in this study based on our group's work on SM [4], and prior studies of neurodevelopment in kidney disease [22]. Behavioral assessments were conducted according to standard operating procedures by trained research assistants with a Bachelor's degrees in psychology. Testing was subjected to a continuous quality control program by a quality control officer that provided feedback on tests to ensure standardization. Children <6 years were tested with preschool instruments and children 6 years or older were tested with school-aged instruments. We evaluated socio-emotional function with the preschool (18 months-5 years, 11 months) and school-aged (6–12 years) Child Behavior Checklist (CBCL) [23], and executive function with the preschool (2–6 years) and school-aged (6–12 years) Behavior Rating Inventory of Executive Function (BRIEF) [24]. Higher z-scores on

these tools indicate poorer performance and more problematic behavior. Children with SM were evaluated one week after hospital discharge and CC were evaluated at enrollment. We repeated testing at 6, 12 months, and 24 months follow-up for all participants. Children who turned 6 during the study were therefore tested with the preschool instruments initially and then with the school-aged instruments. Primary outcomes included internalizing and externalizing behavior (CBCL) and executive function and emergent metacognition (BRIEF) for all ages; and for children <6 the BRIEF-P flexibility index and inhibitory self-control index, and for children ≥6, the BRIEF behavioral regulation index. Secondary outcomes were the CBCL sub-scales for internalizing and externalizing behavior and the BRIEF sub-scales.

## Statistical analysis

All analyses were performed with Stata/SE 14.2 (StataCorp LP). We compared baseline demographic and clinical characteristics based on AKI status using Student's t-test for continuous measures, and Pearson's χ2 test or Fisher's exact test for categorical variables as appropriate. Age-adjusted z-scores for the HOME and all behavioral outcomes were created based on the scores from CC as previously described [4]. Linear mixed effects models were used to compare longitudinal behavioral outcomes among children with SM based on AKI status. All linear mixed models were fitted with a subject specific random intercept and visit as a categorical variable to allow for non-linearity between visits. Behavioral models for children <6 years included a caretaker random effect. In all models, we controlled for covariates selected *a priori* for their reported or hypothesized impact on neurodevelopment: age, sex, weight-for-age z-score, height-for-age z-score, SES total score, HOME z-score, maternal education level, preschool exposure, presence of coma, number of seizures during hospitalization, study year of enrollment, and parenteral anti-malarial therapy (quinine vs. artemisinin), and test administrator [3–5, 25]. Using Holm's procedure, we adjusted for the number of comparisons within each age group (<6 years and ≥6 years) for primary outcomes (children <6 years, n = 6; children ≥6 years, n = 5) and secondary outcomes (children <6 years, n = 12; children ≥6 years, n = 14).

## Ethics and consent

Written informed consent was obtained from the primary caregivers (parents or guardians), and assent in children older than 8 years of age whose primary caregivers consented to participation. This study was approved by the Uganda National Council for Science and Technology, and Institutional Review Boards from Makerere University (School of Medicine Research and Ethics Committee), and the University of Minnesota.

## Results

A total of 440 children with SM (CM, n = 226, or SMA, n = 214) and 173 CC were included in our analysis (Fig 1, Table 1). The prevalence of AKI in the study cohort was 35.1% overall [14] and 33.2% among surviving children with baseline behavioral testing. Overall children with SM were comparable in age to the CC with a mean age (SD) of 4.00 (2.03) years in CC compared to 3.71 (1.84) years of age in children with SM. A lower proportion of children with SM were female compared to CC (40.2% vs. 54.3% respectively, p = 0.002). At enrollment, 51 (29.5%) of the CC had asymptomatic malaria detected by PCR. The presence of asymptomatic parasitemia in the CC was not associated with differences in our primary behavioral outcomes (S1 Table).

Table 1 summarizes the baseline demographic and clinical characteristics of children with severe malaria, with and without AKI on enrollment. Children with AKI had higher mean

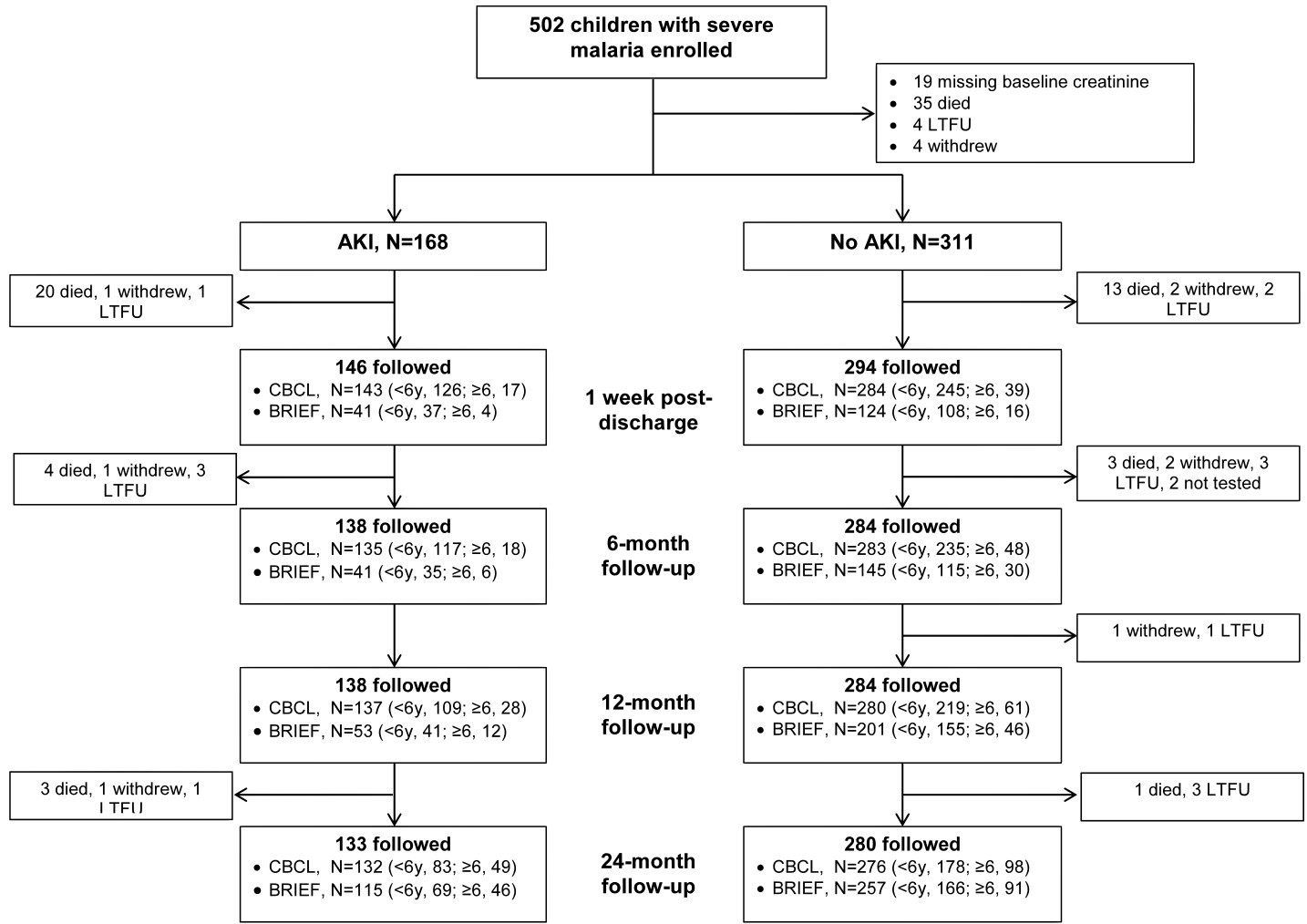

**Fig 1. Flow chart of the study population.** Abbreviations, acute kidney injury (AKI), lost-to-follow-up (LTFU), child behavior checklist (CBCL), Behavior Rating Inventory of Executive Function (BRIEF).

height-for-age z-scores (p = 0.037), lower socioeconomic status (p = 0.003), and lower mean home environment quality z scores (p = 0.005) than children without AKI. Children with AKI reported a shorter duration of fever prior presentation suggesting that AKI was not related to prolonged duration of illness (p = 0.03). Children with AKI were also more likely to be in coma (i.e., have CM, p<0.001) and had a higher mean number of seizures during admission (p<0.001) than children without AKI. Most children presented with a BUN-to-creatinine ratio suggestive of prerenal injury with 92.2% of children without AKI and 95.2% of children with AKI having a BUN-to-creatinine ratio >20. Repeated creatinine measurements were not available for the children, and creatinine was measured retrospectively so AKI status was not known at the time of hospitalization.

## Longitudinal behavioral outcomes among children with severe malaria and acute kidney injury

Children ≥6 years with AKI exhibited poorer externalizing behavior than those without AKI (Beta (95% CI), 0.52 (0.20, 0.85), p = 0.001) (Table 2), characterized by more aggression (Beta

**Table 1. Baseline characteristics of children with severe malaria by AKI status.**

| | Community children N = 173 | Severe Malaria | | |
| --- | --- | --- | --- | --- |
| | | No AKI (n = 294) | AKI (n = 146) | P value No AKI vs. AKI |
| Age in years | 4.00 ± 2.03 | 3.74 ± 1.84 | 3.65 ± 1.84 | 0.64 |
| Duration of illness, days | N/A | 4.1 ± 3.0 | 3.5 ± 1.9 | 0.03 |
| Female sex, No. (%) | 94 (54.3) | 119 (40.5) | 58 (39.7) | 0.88 |
| Weight-for-age z-score | -0.95 ± 0.98 | -1.23 ± 1.11 | -1.40 ± 1.00 | 0.13 |
| Height-for-age z-score | -1.05 ±1.14 | -1.15 ± 1.46 | -0.86 ± 1.35 | 0.037 |
| Weight-for-height z-score | -0.34 ± 1.04 | -0.87 ± 1.34 | -0.89 ± 1.25 | 0.86 |
| Socioeconomic status (SES) | 9.92 ± 3.03 | 9.90 ± 3.23 | 8.92 ± 2.89 | 0.003 |
| Home environment z score | 0.11 ± 1.03 | 0.08 ± 1.00 | -0.20 ± 1.01 | 0.005 |
| Maternal education level, No. (%) | | | | |
| Primary 6 or lower | 47 (27.2) | 108 (36.7) | 60 (41.1) | 0.34 |
| Primary 7 | 42 (24.3) | 61 (20.8) | 31 (21.2) | |
| Secondary or higher | 74 (42.8) | 116 (39.5) | 47 (32.2) | |
| Not known | 10 (5.8) | 9 (3.1) | 8 (5.5) | |
| Paternal education level, No. (%) | | | | |
| Primary 6 or lower | 24 (23.9) | 63 (21.4) | 25 (17.1) | 0.54 |
| Primary 7 | 33 (19.1) | 43 (14.6) | 28 (19.2) | |
| Secondary or higher | 90 (52.0) | 131 (44.6) | 65 (44.5) | |
| Not known | 26 (15.0) | 57 (19.4) | 28 (19.2) | |
| Preschool education, No. (%) | 72 (41.9) | 102 (35.3) | 39 (27.1) | 0.09 |
| HIV positive at baseline, No. (%) | 3 (1.2) | 6 (2.1) | 5 (3.5) | 0.28 |
| Hemoglobin SS genotype, No. (%) | 0 (0.0) | 16 (5.4) | 4 (2.8) | 0.10 |
| Hypotensive on admission[a], No. (%) | --- | 0 (0.0) | 1 (0.7) | 0.16 |
| Hypoglycemia on admission, No. (%) | --- | 12 (4.1) | 8 (5.5) | 0.28 |
| Severe anemia on admission, No. (%) | --- | 178 (60.5) | 86 (58.9) | 0.74 |
| Coma on admission, No. (%) | --- | 134 (45.6) | 92 (63.0) | <0.001 |
| No. seizures during hospitalization | --- | 0.38 ± 1.00 | 1.11 ± 2.51 | <0.001 |

**Abbreviations**: N/A, not applicable; P, P-value; HIV, Human Immunodeficiency Virus; AKI, acute kidney injury

Data presented as mean ± standard deviation or median (interquartile range)

Continuous measures compared with Student's t-test (mean, standard deviation) or Wilcoxon Rank sum test (median, interquartile range). Categorical measures compared with Pearson's $X^2$ or Fisher's exact test (for any comparison in which any cell count <5).

[a]Hypotension defined as systolic blood pressure < 50 mmHg children <5 years, <70 mmHg children <12 years

(95% CI), 0.49 (0.18, 0.81), p = 0.001) (Table 3). The executive functioning (global executive composite) of children ≥6 years with AKI was also worse compared with those without AKI (Beta (95% CI), 0.48 (0.15, 0.82), p = 0.005) (Table 2). Their caregivers reported poorer behavioral regulation (Beta (95% CI), 0.66 (0.32, 1.01), p = 0.0002), as evidenced by poorer impulse control (Beta (95% CI), 0.64 (0.30, 0.97), p = 0.0002) and emotional control (Beta (95% CI), 0.60 (0.28, 0.93), p = 0.0003) (Table 3). All models adjusted for child age, sex, nutritional status (height-for-age and weight-for-age z scores), socioeconomic status, enrichment in the home environment, preschool exposure, neurological complications including coma on admission and number of seizures during hospitalization, parenteral antimalarial treatment (quinine vs. artemisinin), year of enrollment and test administrator.

To determine whether the relationship between AKI and behavioral differences reflected increased disease severity rather than a relationship between AKI *per se*, we considered models where we adjusted for disease severity using the Lambaréné Organ Dysfunction Score (LODS)

**Table 2. Primary behavioral outcomes for children with severe malaria by acute kidney injury (AKI) status.**

| | Unadjusted Estimates | | | Adjusted Estimates | | |
|---|---|---|---|---|---|---|
| | N (obs.), N | Coefficient (95% CI) | P | N (obs.), N | Coefficient (95% CI) | P |
| **Children aged <6 years** | | | | | | |
| **Socio-emotional function[a]** | | | | | | |
| Internalizing behavior | 1296, 379 | 0.09 (-0.07, 0.26) | 0.28 | 1276, 370 | -0.08 (-0.25, 0.07) | 0.27 |
| Externalizing behavior | 1296, 379 | 0.19 (0.02, 0.37) | 0.03 | 1276, 370 | 0.06 (-0.12, 0.24) | 0.49 |
| **Executive function[b]** | | | | | | |
| Global Executive Composite | 726, 298 | 0.41 (0.13, 0.69) | 0.004 | 571, 283 | 0.04 (-0.24, 0.32) | 0.76 |
| Emergent Metacognition Index | 726, 298 | 0.48 (0.20, 0.76) | 0.0008 | 571, 283 | 0.14 (-0.14, 0.43) | 0.32 |
| Flexibility Index | 726, 298 | 0.24 (0.003, 0.48) | 0.05 | 571, 283 | -0.06 (-0.30, 0.17) | 0.60 |
| Inhibitory Self-Control Index | 726, 298 | 0.27 (0.005, 0.54) | 0.05 | 571, 283 | -0.08 (-0.37, 0.21) | 0.58 |
| **Children aged ≥6 years** | | | | | | |
| **Socio-emotional function[a]** | | | | | | |
| Internalizing behavior | 358, 150 | 0.11 (-0.12, 0.34) | 0.35 | 357, 149 | -0.01 (-0.23, 0.22) | 0.95 |
| Externalizing behavior | 358, 150 | 0.58 (0.25, 0.90) | 0.0006 | **357, 149** | **0.52 (0.20, 0.85)** | **0.002†** |
| **Executive function[b]** | | | | | | |
| Global Executive Composite | 251, 137 | 0.43 (0.09, 0.77) | 0.01 | **251, 137** | **0.48 (0.15, 0.82)** | **0.005†** |
| Emergent Metacognition Index | 251, 137 | 0.30 (-0.04, 0.65) | 0.08 | 251, 137 | 0.27 (-0.08, 0.62) | 0.13 |
| Behavior Regulation Index | 251, 137 | 0.52 (0.17, 0.88) | 0.004 | 251, 137 | **0.66 (0.32, 1.01)** | **0.0002†** |

Abbreviations: CI, confidence interval; P, P-value; N (obs.), number of observations in the model; N, the number of children in the analysis.

[a]Assessed using the Child Behavior Checklist (CBCL)

[b]Assessed using the Behavior Rating Inventory of Executive Function (BRIEF)

All linear mixed models were fitted with a subject specific random intercept and a caretaker random effect (for children <6 years of age) and visit as a categorical variable (baseline, 6 months, 12 months, 24 months).

Adjusted models included age, sex, height-for-age, weight-for-age, socioeconomic status, home environment, maternal education, preschool exposure, presence of coma on admission, number of seizures during hospitalization, parenteral antimalarial treatment (quinine vs. artemisinin), year of enrollment and test administrator as fixed effects.

† Significant following Holm's adjustment for multiple comparisons within each age group (children <6, n = 6; children ≥6, n = 5).

[26, 27], a validated prognostic score reflecting disease severity in malaria. We also considered models where other individual SM complications were included: jaundice, lactic acidosis (Table 4). In each instance, inclusion of the LODS or other SM complications did not affect the relationship between AKI and behavioral differences. We further considered whether differences in fluid resuscitation or transfusion in AKI [28] affected behavioral outcomes, and found no differences in outcomes based on fluid resuscitation or transfusion and the relationship between behavioral differences and AKI remained significant (Table 4).

In contrast to the findings in older children, there were no behavioral differences children <6 years in primary (Table 2) or secondary outcomes (S2 Table) based on AKI status at admission. The longitudinal mixed effects model includes data from all baseline and follow-up assessments, so children who had assessments done at ≥6 years were not necessarily ≥6 years at the time of their severe malaria episode. 63% of the assessments done at ≥6 years were in children who were <6 years at the time of their enrollment SM episode. Effect sizes of differences in externalizing behavior, executive function and behavioral regulation were similar for these children as for those ≥6 years at initial testing (S3 Table).

To determine whether the relationship between AKI and behavior was consistent in children with CM and SMA, we performed a stratified analysis by severe malaria group. The

**Table 3. Secondary socio-emotional outcomes for children ≥6 years with severe malaria by acute kidney injury (AKI) status.**

| | Unadjusted models | | | Adjusted Models | | |
|---|---|---|---|---|---|---|
| | N (obs.), N | Coefficient (95% CI) | P | N (obs.), N | Coefficient (95% CI) | P |
| **Socio-emotional function[a]** | | | | | | |
| **Internalizing scales** | | | | | | |
| Anxious/depressed | 358, 150 | 0.06 (-0.19, 0.31) | 0.64 | 357, 149 | 0.01 (-0.24, 0.27) | 0.93 |
| Withdraw/depressed | 358, 150 | 0.16 (-0.11, 0.44) | 0.25 | 357, 149 | 0.05 (-0.22, 0.32) | 0.71 |
| Somatic complaints | 358, 150 | -0.03 (-0.26, 0.19) | 0.77 | 357, 149 | -0.07 (-0.30, 0.16) | 0.55 |
| **Externalizing scales** | | | | | | |
| Aggressive behavior | 358, 150 | 0.54 (0.21, 0.86) | 0.001 | 357, 149 | **0.49 (0.18, 0.81)** | **0.003†** |
| Rule breaking | 358, 150 | 0.38 (0.11, 0.66) | 0.007 | 357, 149 | 0.32 (0.03, 0.61) | 0.03 |
| Social problems | 358, 150 | 0.45 90.12, 0.78) | 0.007 | 357, 149 | 0.32 (-0.02, 0.65) | 0.06 |
| Thought problems | 358, 150 | 0.14 (-0.13, 0.41) | 0.31 | 357, 149 | 0.07 (-0.21, 0.36) | 0.61 |
| Attention problems | 358, 150 | 0.20 (-0.19, 0.59) | 0.32 | 357, 149 | 0.16 (-0.22, 0.53) | 0.41 |
| **Executive function[b]** | | | | | | |
| **Behavior Regulation Index** | | | | | | |
| Inhibit | 251, 137 | 0.50 (0.17, 0.83) | 0.003 | 251, 137 | **0.64 (0.30, 0.97)** | **0.0002†** |
| Task shifting | 251, 137 | 0.20 (-0.14, 0.54) | 0.25 | 251, 137 | 0.31 (-0.03, 0.65) | 0.07 |
| Emotional control | 251, 137 | 0.52 (0.19, 0.84) | 0.002 | 251, 137 | **0.60 (0.28, 0.93)** | **0.0003†** |
| **Metacognition Index** | | | | | | |
| Initiate | 251, 137 | 0.27 (-0.05, 0.59) | 0.10 | 251, 137 | 0.31 (-0.01, 0.62) | 0.05 |
| Plan/organize | 251, 137 | 0.10 (-0.20, 0.40) | 0.52 | 251, 137 | 0.07 (-0.23, 0.36) | 0.66 |
| Organization of materials | 251, 137 | 0.07 (-0.28, 0.42) | 0.69 | 251, 137 | -0.06 (-0.43, 0.31) | 0.74 |
| Monitor | 251, 137 | 0.38 (0.06, 0.71) | 0.02 | 251, 137 | 0.32 (-0.04, 0.69) | 0.08 |

**Abbreviations:** CI, confidence interval; P, P-value; N (obs.), number of observations in the model; N, the number of children in the analysis.

[a]Assessed using the Child Behavior Checklist (CBCL)

[b]Assessed using the Behavior Rating Inventory of Executive Function (BRIEF)

All linear mixed models were fitted with a subject specific random intercept and visit as a categorical variable (baseline, 6 months, 12 months, 24 months).

Adjusted models included age, sex, height-for-age, weight-for-age, socioeconomic status, home environment, maternal education, preschool exposure, presence of coma on admission, number of seizures during hospitalization, parenteral antimalarial treatment (quinine vs. artemisinin), year of enrollment and test administrator as fixed effects.

† Significant following Holm's adjustment for multiple comparisons (n = 14).

association with diminished executive function and AKI remained significant for children with both cerebral malaria and severe malaria (S4 Table). AKI remained an independent risk factor for externalizing behavior in children ≥6 years of age with CM.

## Discussion

AKI is increasingly being recognized as an important complication in children hospitalized with severe malaria and is an independent risk factor for long-term neurocognitive deficits in survivors [14]. We have previously shown that children with CM or SMA had worse behavioral outcomes compared to CC [25]. In this study, we explored whether the presence of AKI on admission was related to the behavioral differences previously observed. Among children with CM or SMA, AKI was associated with behavioral problems, notably externalizing behavior, poor behavioral regulation, and executive function (global executive composite). This effect was independent of other measures of disease severity and known risk factors for worse

**Table 4. Behavioral outcomes adjusting for severe malaria complications and treatment.**

| | Children ≥6 years | | |
|---|---|---|---|
| | N (obs.), N | Coefficient (95% CI) | *P* |
| **Adjusting for disease severity** | | | |
| **Socio-emotional function[a]** | | | |
| Internalizing behavior | 357, 149 | 0.01 (-0.21, 0.24) | 0.919 |
| Externalizing behavior | 357, 149 | **0.55 (0.22, 0.87)** | **0.001** |
| **Executive function[b]** | | | |
| Global Executive Composite | 251, 137 | **0.48 (0.14, 0.82)** | **0.006** |
| Emergent Metacognition Index | 251, 137 | 0.27 (-0.09, 0.62) | 0.139 |
| Behavior Regulation Index | 251, 137 | **0.67 (0.32, 1.01)** | **0.0002** |
| **Adjusting for acidosis** | | | |
| **Socio-emotional function[a]** | | | |
| Internalizing behavior | 334, 139 | -0.06 (-0.28, 0.16) | 0.605 |
| Externalizing behavior | 334, 139 | **0.51 (0.18, 0.85)** | **0.003** |
| **Executive function[b]** | | | |
| Global Executive Composite | 228, 127 | **0.45 (0.09, 0.81)** | **0.016** |
| Emergent Metacognition Index | 228, 127 | 0.23 (-0.15, 0.62) | 0.225 |
| Behavior Regulation Index | 228, 127 | **0.64 (0.27, 1.01)** | **0.0008** |
| **Adjusting for jaundice** | | | |
| **Socio-emotional function[a]** | | | |
| Internalizing behavior | 357, 149 | 0.01 (-0.22, 0.23) | 0.957 |
| Externalizing behavior | 357, 149 | **0.51 (0.18, 0.83)** | **0.003** |
| **Executive function[b]** | | | |
| Global Executive Composite | 251, 137 | **0.45 (0.12, 0.79)** | **0.009** |
| Emergent Metacognition Index | 251, 137 | 0.23 (-0.12, 0.59) | 0.190 |
| Behavior Regulation Index | 251, 137 | **0.64 (0.30, 0.99)** | **0.0003** |
| **Adjusting for fluid resuscitation and transfusion** | | | |
| **Socio-emotional function[a]** | | | |
| Internalizing behavior | 357, 149 | -0.01 (-0.23, 0.22) | 0.957 |
| Externalizing behavior | 357, 149 | **0.54 (0.22, 0.87)** | **0.001** |
| **Executive function[b]** | | | |
| Global Executive Composite | 251, 137 | **0.48 (0.14, 0.82)** | **0.006** |
| Emergent Metacognition Index | 251, 137 | 0.26 (-0.10, 0.62) | 0.151 |
| Behavior Regulation Index | 251, 137 | **0.68 (0.34, 1.02)** | **0.0001** |

**Abbreviations:** CI, confidence interval; P, P-value; N (obs.), number of observations in the model; N, the number of children in the analysis.

[a]Assessed using the Child Behavior Checklist (CBCL)

[b]Assessed using the Behavior Rating Inventory of Executive Function (BRIEF)

All linear mixed models were fitted with a subject specific random intercept and visit as a categorical variable (baseline, 6 months, 12 months, 24 months).

Adjusted models included age, sex, height-for-age, weight-for-age, socioeconomic status, home environment, maternal education, preschool exposure, number of seizures during hospitalization (in cerebral malaria), parenteral antimalarial treatment (quinine vs. artemisinin), year of enrollment and test administrator as fixed effects. As coma is part of LODS, the presence of coma was not included in the model with LODS.

behavioral outcomes. The long-term behavioral changes associated with AKI in children with SM were evident in school-age children (≥6 years of age) irrespective of the age at insult.

Our finding that children ≥6 years with AKI in SM exhibited more externalizing (aggressive, rule-breaking) behavior, were more impulsive, and had poorer control over their emotional responses is consistent with our findings that AKI is also associated with neurocognitive

impairment in children following SM [7, 14]. Developmentally disabled and otherwise cognitive impaired children struggle more than non-impaired children with behavioral auto-regulation, and can be more prone to behavioral escalation [29, 30]. That we did not find the same behavioral differences in children <6 years with AKI may be due in part to recall bias introduced by our caregiver-report measures. More is expected socially and behaviorally of older children, and undesirable behaviors might have been more frequently recalled by caregivers of children ≥6 years as compared with those <6 years. In addition, the neuro-architecture of infants and toddlers is more plastic than that of school-aged children [31], and thus the younger children in our cohort may have recovered better from the initial developmental insult. However, when stratifying children based their age at enrollment, the effect sizes for differences in externalizing behavior, executive function and behavioral regulation were similar for children <6 years of age as those ≥6 years at initial testing (S3 Table), suggesting that differences associated with AKI were related to the ability to elicit specific behaviors in older children, rather than to AKI-associated behavioral problems occurring only in children with SM ≥6 years of age. We previously reported that both cerebral malaria and severe malarial anemia are associated with poorer developmental outcomes in children <5 years, when compared with CC [25]. The current analysis, however, compares outcomes associated with a specific severe malaria complication among children with severe malaria on enrollment, which may explain the difference in impacted age groups.

We identified developmental impacts that persisted 2 years after SM infection. This persistence may be a function of the pathophysiology of AKI in SM, but should also be contextualized as part of the impacted children's developmental milieu. Atypically developing children interact differently with caregivers, who, in turn, interact differently with them, potentially leading to further developmental disparities [32]. The families of cognitively impaired children in resource- and education-limited settings like Uganda often experience insufficient support services and negative socio-cultural attitudes that may impair recovery from an acute developmental insult [33, 34].

Although AKI was more common among children with coma (CM) than those without (SMA) in our cohort of children with these two forms of SM, the behavioral changes in children with AKI were significant after controlling for the presence of coma and number of seizures during hospitalization. In addition, the relationship between AKI and behavior remained unchanged in models when adjusting for other SM complications, disease severity, or fluid resuscitation and transfusion (Table 4). AKI may therefore be an independent contributor to the well-documented developmental consequences of SM and may explain, at least in part, why children without neurologic complications on admission are at risk for long-term cognitive and behavioral problems.

Microvascular obstruction and endothelial activation have been implicated in both neurologic manifestations of SM [35–37], and the pathogenesis of malaria-associated AKI [37, 38]. AKI has been associated with endothelial activation and blood-brain-barrier integrity in various inflammatory models including sepsis, liver failure, and neurologic disease (reviewed in [39]). Kidney injury itself may lead to systemic inflammation and endothelial activation [39, 40], but can also slow the clearance of inflammatory and neurotoxic metabolites thus exacerbating inflammatory responses [41].

The design of this study enabled us to track neurodevelopment long-term, allowing us to explore what happens after discharge, the primary endpoint for previous studies of AKI and development [7, 42]. Our model took into consideration social and clinical factors that might also have explained the cognitive and behavioral effects we observed. This study benefited from the inclusion of healthy, age-matched community children who were used to derive population-specific kidney function and neurodevelopmental performance curves, allowing us to define impairment in a way that avoided comparisons with children from different social and

economic contexts. Additional studies are needed to determine whether long-term neurocognitive and behavior problems associated with AKI are specific to severe malaria or generalizable to other populations with critical illness. Further, as AKI is a risk factor for subsequent development of chronic kidney disease in children [28, 43, 44], follow-up studies are needed to delineate the long-term impact of chronic kidney disease on child development in survivors.

While caregiver-report behavioral measures are subject to recall and reporter bias, both the BRIEF and CBCL were validated against tester/observer-based measures of children's behavior [23, 24], and have been used successfully in Uganda before [45]. For children <6 years, we included a random caregiver effect to help address reporter bias. Asymptomatic malaria infection, per admission PCR, was present in almost 30% of our community children. Asymptomatic parasitemia was not associated with worse behavioral outcomes (S3 Table), and given the high prevalence of asymptomatic malaria in Ugandan children, we believe that these community children did represent "typical" development for this demographic [18].

## Conclusion

Our study demonstrates that AKI is associated with poorer long-term socio-emotional and executive function in children with SM, independent of sociodemographic factors, neurologic risk factors (e.g. presence of coma, number of seizures during hospitalization), and other SM complications. Our findings suggest that prevention or early management of AKI in these children could mitigate the pervasive, long-term developmental consequences associated with SM, and that expanding access to renal replacement therapy may help children both clinically and developmentally. In endemic regions, however, where both acute care and developmental services are extremely limited, eradication remains the most powerful way to protect children from both the acute and chronic consequences of malaria.

## Supporting information

**S1 Data. AKI Behavior Data Export.** File containing primary data used in the analysis in wide and long format, and an accompanying codebook.
(XLS)

**S1 Table. Primary behavioral outcomes for community children based on the presence of malaria by PCR on enrollment.**
(DOCX)

**S2 Table. Secondary socio-emotional outcomes for children <6 years with severe malaria by acute kidney injury (AKI) status.**
(DOCX)

**S3 Table. Behavioral outcomes for children ≥6 years with severe malaria by acute kidney injury (AKI) status stratified by age at enrollment.**
(DOCX)

**S4 Table. Behavioral outcomes for children ≥6 years with severe malaria by acute kidney injury (AKI) status stratified by study group at enrollment.**
(DOCX)

## Acknowledgments

We thank the children and their parents who participated in this study, the study team for their dedicated effort in treating the children and collecting the data.

## Author Contributions

**Conceptualization:** Meredith R. Hickson, Andrea L. Conroy, Paul Bangirana, Robert O. Opoka, Richard Idro, John M. Ssenkusu, Chandy C. John.

**Formal analysis:** Meredith R. Hickson, Andrea L. Conroy, John M. Ssenkusu.

**Funding acquisition:** Richard Idro, Chandy C. John.

**Investigation:** Paul Bangirana, Robert O. Opoka, Chandy C. John.

**Methodology:** Andrea L. Conroy, Paul Bangirana, Richard Idro, John M. Ssenkusu, Chandy C. John.

**Resources:** Robert O. Opoka, Richard Idro.

**Supervision:** Andrea L. Conroy, Paul Bangirana, Robert O. Opoka, Richard Idro, Chandy C. John.

**Validation:** John M. Ssenkusu.

**Writing – original draft:** Meredith R. Hickson.

**Writing – review & editing:** Andrea L. Conroy, Paul Bangirana, Robert O. Opoka, Richard Idro, John M. Ssenkusu, Chandy C. John.

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
