## [Decision Letter · Decision Letter 0]

30 Oct 2019

PONE-D-19-26285

Acute kidney injury in Ugandan children with severe malaria is associated with long-term behavioral problems

PLOS ONE

Dear Dr Conroy,

Thank you for submitting your manuscript to PLOS ONE. After careful consideration, we feel that it has merit but does not fully meet PLOS ONE’s publication criteria as it currently stands. Therefore, we invite you to submit a revised version of the manuscript that addresses the points raised during the review process.

Both reviewers are highly supportive of publishing this study and point out the current interest in the role of kidney injury in paediatric malaria.  They both have some very minor comments that you might wish to address, but I do not envisage having to send the manuscript out for re-review once you have done this.  

We would appreciate receiving your revised manuscript by Dec 14 2019 11:59PM. To enhance the reproducibility of your results, we recommend that if applicable you deposit your laboratory protocols in protocols.io, where a protocol can be assigned its own identifier (DOI) such that it can be cited independently in the future. For instructions see: http://journals.plos.org/plosone/s/submission-guidelines#loc-laboratory-protocols

We look forward to receiving your revised manuscript.

Kind regards,

Alister G. Craig

Academic Editor

PLOS ONE

Journal Requirements:

Reviewers' comments:

Reviewer's Responses to Questions

**Comments to the Author**

1. Is the manuscript technically sound, and do the data support the conclusions?

Reviewer #1: Yes

Reviewer #2: Yes

2. Has the statistical analysis been performed appropriately and rigorously? 

Reviewer #1: Yes

Reviewer #2: Yes

3. Have the authors made all data underlying the findings in their manuscript fully available?

Reviewer #1: Yes

Reviewer #2: Yes

4. Is the manuscript presented in an intelligible fashion and written in standard English?

Reviewer #1: Yes

Reviewer #2: Yes

5. Review Comments to the Author

Reviewer #1: This is a report assessing the correlation between malaria associated Acute Kidney Injury (AKI) and long-term behavioral problems. The group is a leader in the malaria field and has recently clarified and highlighted the previously underrecognized role of kidney injury in pediatric malaria. The report uses a cohort of malaria patients in Uganda that have been used for multiple publications, including a recent report in BMC Medicine showing the effect of AKI on short- and long-term neuro cognitive outcomes. Using the same cohort, the investigators use the CBCL and BRIEF behavioral assessment tools for two years post discharge. They have found significant deficits in the areas of externalizing behavior, global executive function, and behavior regulation in those children with malaria-related AKI. These findings remain significant in a mixed model that includes twelve covariates as fixed effects.

The study is well performed, the data analyzed appropriately and the results clear. The problem is a large one – with malaria related morbidity having an immense impact across sub-Saharan Africa. I would recommend publication with only the following small comments.

1) I am interested in the role of AKI itself (as opposed to AKI specifically in the setting of malaria) in these long-term sequelae. Are there any data on the role of AKI of other etiologies on long-term neurocognitive or behavioral changes?

2) The conclusion sentence may be confusing for those without a knowledge of the definition of ‘school aged children’. Perhaps the sentence should explicitly state the cut-off of 6 years of age.

3) Very minor. There appear to be some typos in the sentences starting on lines 115 and 277.

Reviewer #2: This is a well conducted study in a fairly large number of African children which has expanded on the authors previous studies which evaluated the association between AKI and neurological disability and cognitive impairment

The authors have now detailed the impact of CM/SM and AKI on behavioral issues with longitudinal assessment. This finding done in a robust and standardized method is novel data.

There are only a few minor comments

1) Was creatinine only assessed at enrollment? How did the authors assess if impaired renal function at the time of assessment at months 12 and 24 months contribute to their results

2) It will be good to get more details on the who did the behavioral assessments and how were they trained to ensure standardization

6. PLOS authors have the option to publish the peer review history of their article (what does this mean?). If published, this will include your full peer review and any attached files.

Reviewer #1: No

Reviewer #2: Yes: Tsin Wen Yeo

---

## [Author Response · Author response to Decision Letter 0]

5 Nov 2019

We have responded to all revision requests to the best of our abilities, and have uploaded a response to reviewers that details all the edits to the manuscript.

---

## [Editor Report · Decision Letter 1]

27 Nov 2019

Acute kidney injury in Ugandan children with severe malaria is associated with long-term behavioral problems

PONE-D-19-26285R1

Dear Dr. Conroy,

We are pleased to inform you that your manuscript has been judged scientifically suitable for publication and will be formally accepted for publication once it complies with all outstanding technical requirements.

With kind regards,

Alister G. Craig

Academic Editor

PLOS ONE

Additional Editor Comments (optional):

Reviewers' comments: Thank you for responding fully to the reviewers' comments.

---

## [Editor Report · Acceptance letter]

10 Dec 2019

PONE-D-19-26285R1 

Acute kidney injury in Ugandan children with severe malaria is associated with long-term behavioral problems 

Dear Dr. Conroy:

I am pleased to inform you that your manuscript has been deemed suitable for publication in PLOS ONE. Congratulations! Your manuscript is now with our production department. 

With kind regards,

on behalf of

Prof. Alister G. Craig 

Academic Editor

PLOS ONE